# A three-order-parameter bistable magnetoelectric multiferroic metal

Andrea Urru[1,2], Francesco Ricci[3], Alessio Filippetti[1,4], Jorge Íñiguez [5,6] & Vincenzo Fiorentini [1✉]

Using first-principles calculations we predict that the layered-perovskite metal $Bi_5Mn_5O_{17}$ is a ferromagnet, ferroelectric, and ferrotoroid which may realize the long sought-after goal of a room-temperature ferromagnetic single-phase multiferroic with large, strongly coupled, primary-order polarization and magnetization. $Bi_5Mn_5O_{17}$ has two nearly energy-degenerate ground states with mutually orthogonal vector order parameters (polarization, magnetization, ferrotoroidicity), which can be rotated globally by switching between ground states. Giant cross-coupling magnetoelectric and magnetotoroidic effects, as well as optical non-reciprocity, are thus expected. Importantly, $Bi_5Mn_5O_{17}$ should be thermodynamically stable in O-rich growth conditions, and hence experimentally accessible.

[1] Dipartimento di Fisica, Università di Cagliari, Cittadella Universitaria, Monserrato, I-09042 Cagliari, Italy. [2] Scuola Superiore Internazionale di Studi Avanzati, Via Bonomea 265, I-34136 Trieste, Italy. [3] Institute of Condensed Matter and Nanosciences (IMCN), Université Catholique de Louvain, Chemin des Étoiles 8, B-1348 Louvain-la-Neuve, Belgium. [4] CNR-IOM, UOS Cagliari, Cittadella Universitaria, Monserrato, I-09042 Cagliari, Italy. [5] Materials Research and Technology Department, Luxembourg Institute of Science and Technology, 5 avenue des Hauts-Fourneaux, L-4362 Esch/Alzette, Luxembourg. [6] Department of Physics and Materials Science, University of Luxembourg, 41 Rue du Brill, L-4408 Belvaux, Luxembourg. ✉email: vincenzo.fiorentini@gmail.com

One of the key goals of multiferroics research[1], not yet achieved after several decades, is finding a room-temperature single-phase multiferroic with large polarization and magnetization primary orders—i.e. neither being a weak side effect of other phenomena. In this context, a multiferroic metal would be of the utmost interest as the seat of robust magnetism, enabling stable and large magnetization and polarization at application-relevant temperatures, and hence a possible path to the above goal. Also, such a material would quite likely be a ferromagnet, whereas most insulating magnets are antiferromagnetic.

Further, if additional orders[2] such as e.g. ferrotoroidicity were to exist (as they do in appropriate symmetry), the mutual couplings of the various orders (e.g. magnetoelectricity) may be quite out of the ordinary. On the other hand, more than two concurrent orders rarely coexist in a multiferroic, and this is especially true of metals, where multiferroicity itself is already unexpected (ferromagnetism in metals is common, but ferroelectricity is exceedingly rare[3]).

It is therefore against all expectations that in this paper we predict a specific instance of a multiferroic metal as a possible path to the key goal of multiferroicity: the orthorhombic layered-perovskite $Bi_5Mn_5O_{17}$ (BiMO henceforth) is a multi-order-parameter, bistable, magnetoelectric, metallic, room-temperature multiferroic. Indeed, BiMO is a metal possessing three space-orthogonal vector-order parameters: magnetization $M$, polarization $P$, and ferrotoroidal moment $T$, generated by simultaneous time reversal and inversion symmetry breaking; it exists in two nearly energy-degenerate multiferroic ground states, which can be transformed into one another, causing the order-parameter triad to rotate in space; it exhibits giant magnetoelectricity, and potentially other couplings among the three orders, including toroidicity-related optical effects; finally, it has a sizable thermodynamic stability window, so it can be grown in practice.

## Results

**Structure**. BiMO is a layered perovskite of the class $A_nX_nO_{3n+2}$ with $n = 5$. Its structure is depicted in Fig. 1. The periodic cell comprises two 5-perovskite-unit blocks along the **b** axis (the crystal axes are $a = [100]$, $b = [011]$, $c = [0\bar{1}1]$ in the cubic perovskite setting).

We search for instabilities in the $q = 0$ phonon spectrum of the centrosymmetric structure with $Pmnn$ space group, computed both via density-functional perturbation theory[4] and by finite differences, with completely consistent results. The two dominant unstable modes, with one-dimensional irreps $B_{1u}$ and $B_{3u}$, condense into polar stable ground states with space groups $Pmn2_1$ and $Pm2_1n$.

In both phases (Fig. 1, see also Supplementary Discussion for structural data) the Bi atoms move within the **bc**-plane, and what distinguishes the structures is the modulation of the Bi displacements from layer to layer. A schematic representation, with indicative arrows (not to scale) corresponding to the largest dipoles, is given in Fig. 1. In the $Pmn2_1$ phase (C state in the following), the displacements along **b** form an anti-polar pattern; instead, the displacements along **c** are in phase, originating a **c**-polarized distortion. In the $Pm2_1n$ state, labeled B in the following, an anti-polar pattern appears along **c**; along **b**, an uncompensated anti-polar pattern originates a **b**-polarized distortion. All displacements are invertible and allow for hysteresis; indeed, as discussed below, BiMO supports a depolarizing field. A third unstable mode (irrep $B_{2u}$) leading to a structure with symmetry $P2_1mn$ is preempted by the C and B modes, due to its much lesser energy gain.

The calculated energy gain upon condensing into the C or B state is the same, $100\,\mu eV\,Å^{-3}$, to within 1%. The energy landscape in Fig. 2 shows pictorially the two distinct minima and provides an estimate of about $50\,\mu eV\,Å^{-3}$ for the barrier between the two states. Since this energy barrier is similar to those occurring in other polar perovskites, we can estimate the ferroelectric Curie temperature at well above ambient. More precisely, in prototypical ferroelectric perovskite $BaTiO_3$, equivalent rhombohedral minima are separated by an orthorhombic saddle point, the energy barrier being[5] about $15\,\mu eV\,Å^{-3}$, which results in an orthorhombic–rhombohedral transition temperature of 183 K. Since such a transition temperature is known to scale with the mentioned energy barrier[6,7], and the polar distortion in layered perovskite BiMO is ultimately not very different from those occurring in its perovskite counterparts, we can estimate a Curie temperature for BiMO exceeding 500 K. BiMO will thus be locked in either ground state C or B at room temperature, and

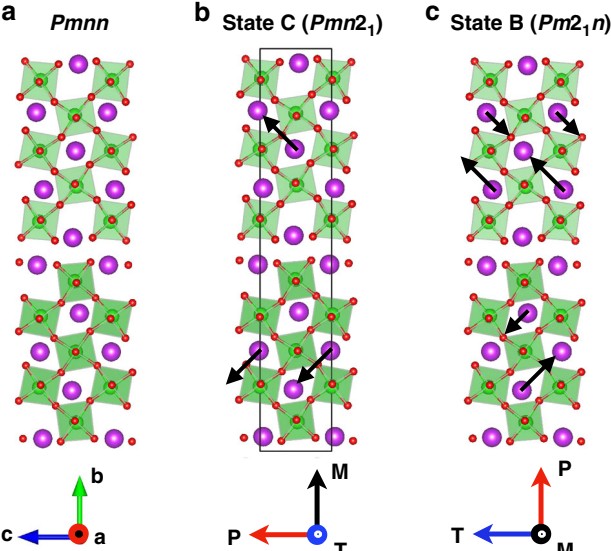

**Fig. 1 Comparison of high-symmetry and distorted structures.** High-symmetry unstable (**a**) and distorted ground-state stable (**b**), (**c**) structures of BiMO. The crystal axes, the main local ionic dipoles, and the order-parameter vectors in the different states are shown.

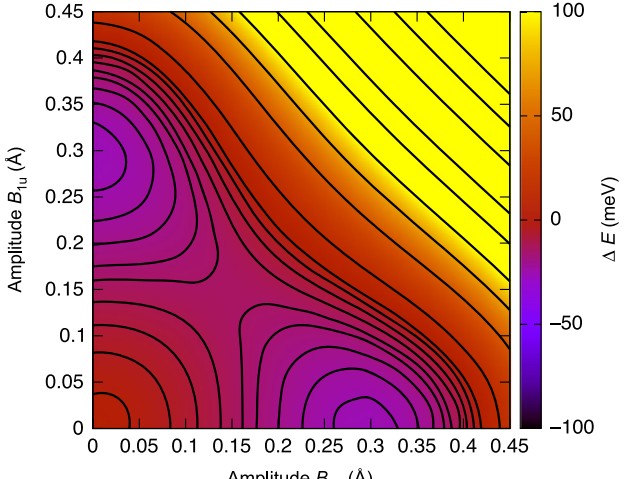

**Fig. 2 Energy landscape for BiMO vs ferroelectric distortions.** Energy of BiMO as a function of the distortion along the two main unstable (and both polar) modes of the high-symmetry phase.

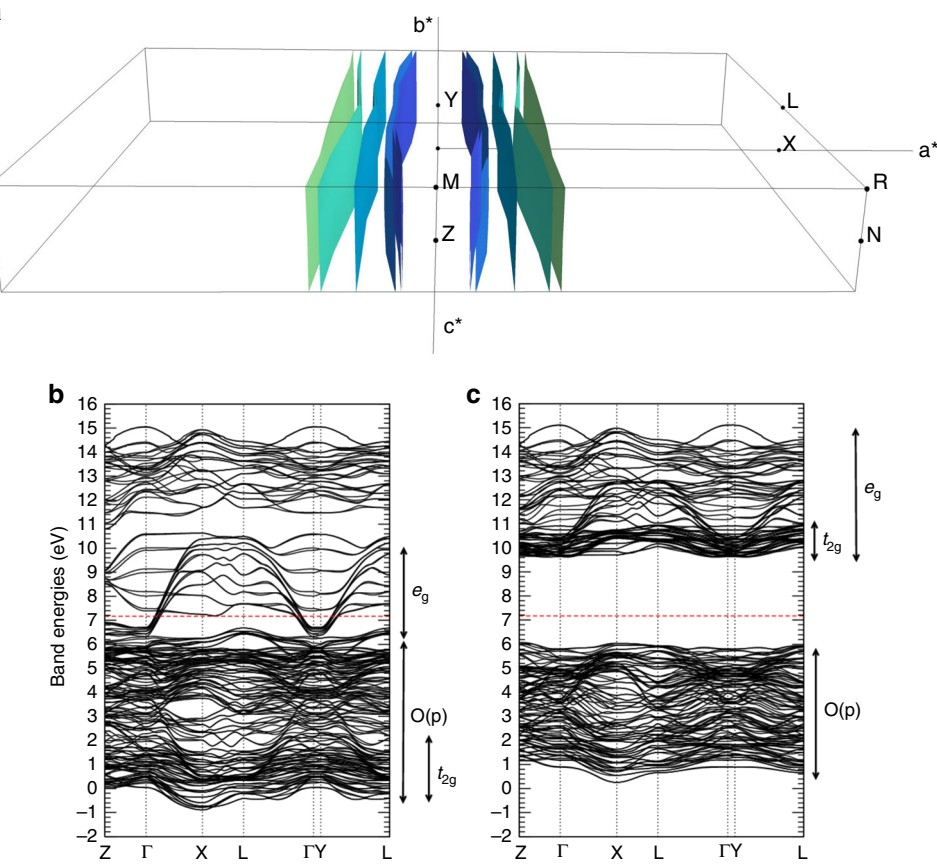

**Fig. 3 Main features of the electronic properties of BiMO. a** 3D view of the BiMO Fermi surface; **b** majority and **c** minority bands of BiMO, calculated with the VPSIC method. Fermi level: horizontal dashed line.

can be thermally activated between them with a modest $T$ increase.

**Polarization**. Given their polar symmetry, both the B and C states of BiMO can possess a spontaneous polarization **P**. The coexistence of metallicity and polarization has been discussed at length in our previous work on the ferroelectric metal $Bi_5Ti_5O_{17}$[3], a layered perovskite to which BiMO bears close similarities. As in that case, we calculate **P** with a modified Berry phase technique[3] which exploits the flatness of the bands along the polar axes and the sheet-like Fermi surface (see below the discussion of the band structure). In the B state, $P_B \| b$ and $|P_B| = 0.71\,\mu C\,cm^{-2}$; in the C state, $P_C \| c$ and $|P_C| = 5.03\,\mu C\,cm^{-2}$. Both values are in the same league as III–V nitrides and II–VI oxides (e.g. $2.9\,\mu C\,cm^{-2}$ for GaN[8]). The electronic polarization contributions by valence majority, valence minority, and conduction electrons are roughly in the ratio 30:10:1.

**Bands and magnetism**. BiMO is a metal, whose Fermi surface (Fig. 3) shows line-like sections along the $b^*$ and (less cleanly) $c^*$ reciprocal axes (i.e. in $b^*c^*$-like planes in the Brillouin zone), justifying the applicability of the approach of ref. [3] to computing the polarization of both states B and C. The bands (computed including self-interaction corrections, which drastically improve predicted gaps[9,10]) show that BiMO is a half-metallic ferromagnet with a large minority gap (GGA results are the same except for the smaller minority gap). This is expected from its nominally $3d^{3.2}$ Mn ions coupled via double exchange[11].

The average magnetization is $3.06\,\mu_B$ per Mn ($2.84\,\mu_B$ per Mn within atomic spheres) in GGA, and $3.4\,\mu_B$ per Mn from VPSIC

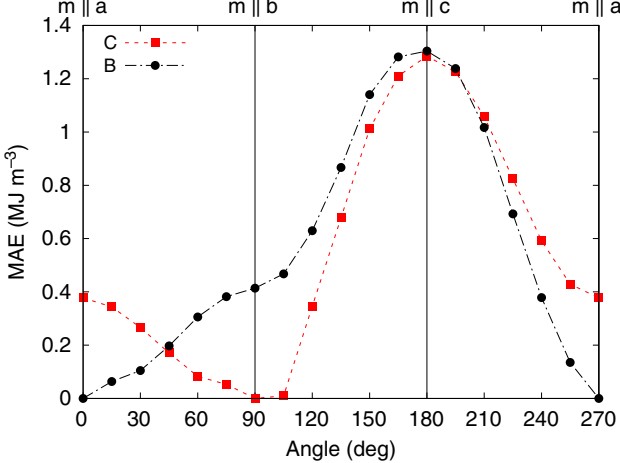

**Fig. 4 Magnetic anisotropy in BiMO.** Energy as a function of the orientation of the magnetization for the C (dashed, squares) and B (dashed–dotted, circles) ground states of BiMO. The easy axis is **b** for the C state, and **a** for the B state.

self-consistent occupations. The energy difference of the ferromagnet and (approximate) G-type antiferromagnet provides an average magnetic coupling $J \simeq 16\,meV$; applying a Hubbard U correction with the typical Mn value $U = 3\,eV$, $J$ becomes about 24 meV (see Supplementary Discussion).

We performed non-collinear spin–orbit calculations to ascertain the orientation of the magnetization. As shown in Fig. 4, in the C state **M** is parallel to the **b** axis, with significant

magnetoanisotropy energy (MAE) barriers of 0.38 MJ m$^{-3}$ for the **a** axis and 1.28 MJ m$^{-3}$ for the **c** axis. For the B state, **M** is instead parallel to the **a** axis (Fig. 4), with magnetoanisotropy barriers 0.41 MJ m$^{-3}$ for the **b** axis and 1.30 MJ m$^{-3}$ for the **c** axis. In both ground states, therefore, **M** is orthogonal to the polar axis and hence to **P** (the polarization was discussed above).

We now provide estimates of the magnetic Curie temperature $T_C^M$ based on literature numerical simulations of the Ising as well as classical and quantum Heisenberg models[12,13]; the magnetoanisotropy energy just calculated is neither especially small or large[14], so it is not obvious which model is preferable. The values of $T_C^M$ for the $J$ mentioned earlier (16 meV for $U = 0$ eV and 24 meV for $U = 3$ eV) are as follows: classical Heisenberg 361 K and 521 K, quantum Heisenberg 263 K and 387 K, Ising 378 K and 555 K. It seems reasonable to conclude that BiMO's $T_C^M$ is near or above room temperature.

**Toroidicity**. The fact that **M** is orthogonal to **P** in both the C and B ground states agrees[2,15,16] with their respective magnetic point groups being $m'm2'$ and $m2'm'$; symmetry further implies[15,16] that there exists a non-zero ferrotoroidal moment **T**[17], the order parameter of a ferrotoroidal state[2,18–22]. **T** is proportional to $\sum_i \mathbf{r}_i \times \mathbf{m}_i$, and is akin to an angular momentum with magnetic moments functioning as velocities; in a toroid, **T** is the sum of individual current-loop terms with **r** = 0 at the center of the torus, hence the name.

Symmetry also implies that the three-order-parameter vectors **M**, **P**, and **T** must be mutually orthogonal. Similarly to polarization, the toroidal moment is defined as a difference between two states[17] (our reference structure has point group $mmm$ and hence zero moment); to obtain a well-defined **T**, one must remove toroidicity quanta analogous to polarization quanta[17], and the cell must be recentered to the average of the magnetic moments positions. Once that is done, we find for the C state **T**∥**a** and $|\mathbf{T}| \simeq 0.27\ \mu_B$Å, and for the B state, **T**∥**c** and $|\mathbf{T}| \simeq 0.77\ \mu_B$Å, smaller than e.g. the 1.75 $\mu_B$Å for LiCoPO$_4$[17], but certainly not insignificant. This establishes **T** as the third order parameter of BiMO, and confirms it to be orthogonal to **P** and **M** in both states C and B, as dictated by symmetry (a similar symmetry-determined situation occurs in other layered perovskites such as[23] V-doped La$_2$Ti$_2$O$_7$ whose magnetic group is 2, and indeed **M**∥**P**∥**T**).

We note in passing that **T** will be non-zero as long as the symmetry is polar, and time reversal is broken; thus **T** would exist even if **P** were suppressed by electronic screening (which it isn't, as discussed below); also, the symmetry of BiMO forbids the existence of the fourth 'electromagnetic' order parameter, the ferroaxial or electrotoroidal moment[20,21].

**Magnetoelectricity and other consequences**. Based on the above results, BiMO should exhibit a number of unique properties and effects. First and foremost, it may undergo multiple magnetoelectric switching, with one order parameter potentially switching under the field conjugate to another order parameter. This can be realized by a trilinear coupling term **T** · (**P** × **M**) in the Landau free-energy expansion (see the Supplementary Discussion): it is both allowed by symmetry[24,25] and consistent with our three orthogonal vector order parameters. (The symmetry of the $Pmnn$ reference structure bars instead **PM**-like bilinear terms.) The trilinear coupling implies that ground states B and C should exist in four distinct degenerate states $1 \equiv (+, +, +)$, $2 \equiv (-, -, +)$, $3 \equiv (-, +, -)$, $4 \equiv (+, -, -)$, where the signs characterize the order parameters in our fixed set of crystallographic axes, and, for example, state $(-, -, +)$ has order parameters $-$**P**, $-$**M**, **T**. We indeed verified directly that these states do exist, with calculations

analogous to those in Fig. 4. It follows that, for example, switching **P** by an electric field in state 1 will lead either to state 2 (magnetization co-switching) or 3 (toroidicity co-switching); and **M** switching by a magnetic field leads state 1 to either 2 or 4. Interestingly, as briefly discussed in the Supplementary Discussion, our calculations suggest that **T** is a secondary (slave) order that follows the primary orders **P** and **M** according to **T** ~ **P** × **M**. Hence, of the mentioned switching possibilities, the ones we expect will occur are 1 → 3 and 1 → 4.

Another class of switching possibilities involves transitions between the two ground states B and C, which entail a space rotation of the vector-order-parameter triad, such as **T**∥**a**, **M**∥**b**, **P**∥**c** ⇒ **M**∥**a**, **P**∥**b**, **T**∥**c** for the C to B transformation. This transition could be driven in several ways: one could electrically pole **P** from **c** to **b**, which should rotate **M** from **b** to **a**, and **T** from **a** to **c**; or more interestingly, a magnetic field coercing **M** from **b** to **a** could turn **P** from **c** to **b**, and **T** from **a** to **c** as well. Since there are four degenerate orientational states in both C and B, there are 16 possible C-to-B transitions: for example, a **b** → **a** magnetization rotation could turn state C $(+, +, +)$ into B $(+, +, +)$, but, due to the degeneracy just discussed, it could also land it into, say, B $(-, +, -)$. It is likely that such transitions will be set apart by different energy barriers, which are however extremely difficult to estimate.

Going further, BiMO should exhibit linear static magnetoelectricity, such as $\delta\mathbf{M} = \widetilde{\alpha}\mathbf{E}$, measured by the magnetoelectric tensor $\widetilde{\alpha}$. According to symmetry, only its off-diagonal elements are non-zero[2], and specifically $\alpha_{bc}$, $\alpha_{cb}$ for state C and $\alpha_{ab}$, $\alpha_{ba}$ for state B (this is similar to the weak-ferromagnet La$_2$Mn$_2$O$_7$[26], whose non-zero tensor elements are $\alpha_{bc}$, $\alpha_{cb}$). Thus, for example, BiMO in state C will exhibit magnetoelectric cross-coupling $\delta M_b = \alpha_{bc}E_c$, so that a **c**-oriented electric field changes the magnetization along **b** (conversely, a **b**-oriented magnetic field would cause a **c**-polarized response). Transforming BiMO to state B, the non-zero elements will be different and so will the cross-coupling, namely $\delta M_a = \alpha_{ab}E_b$, etc. (an important practical consequence of bistability).

Interestingly, due to the trilinear coupling, magnetoelectric coefficients have an antisymmetric component proportional to **T** in addition to the usual symmetric components[17,19,22]. The off-diagonal response also turns out to be related to vanishing of both the ferroaxial moment[2,15,16,22] and the magnetoelectric monopole (which is easily verified to be zero)[27].

Another expected effect in BiMO is optical non-reciprocity, also known as optical-diode effect[28,29] (see also ref. 30 for a review); this is basically tunable and switchable birefringence, visible in magneto-optical absorption[29] and second-harmonic generation (which was used in ref. 19 to establish the ferroic nature of the toroidal order, including hysteretic behavior). It requires non-zero toroidal moment and off-diagonal magnetoelectricity[31], both of which BiMO possesses. Such effects, expected e.g. for beams propagating along opposite directions in a toroidic material, may also occur in BiMO under inversion of **T**, which can be effected via **M** inversion under a magnetic field, e.g. from state 1 to state 4 of a given ground state as described earlier. Additionally, in BiMO a transformation between ground states (C and B) would enable switchable multidirectional birefringence. In passing, we note that the more exotic linear toroidoelectric and toroidomagnetic effects[2] are also possible in this symmetry.

We finally roughly estimate the linear magnetoelectric coupling taking state C as an example. The **P** and **M** changes (with respect to the centrosymmetric phase) $\Delta M = 0.24\ \mu_B$ per cell and $\Delta P = 4.5\ \mu$C cm$^{-2}$ provide a rough estimate of the linear coupling $\alpha = \Delta P/\Delta M = 12\ \mu$s m$^{-1}$, which is large compared to values in boracites[32] or phosphates[33]. One also obtains $\partial M/\partial E \sim \Delta M/\Delta E = \chi_d \alpha$ and $\partial P/\partial H \sim \Delta P/\Delta H = \chi_m \alpha$, equal to, respectively, $3 \times 10^{-5}$

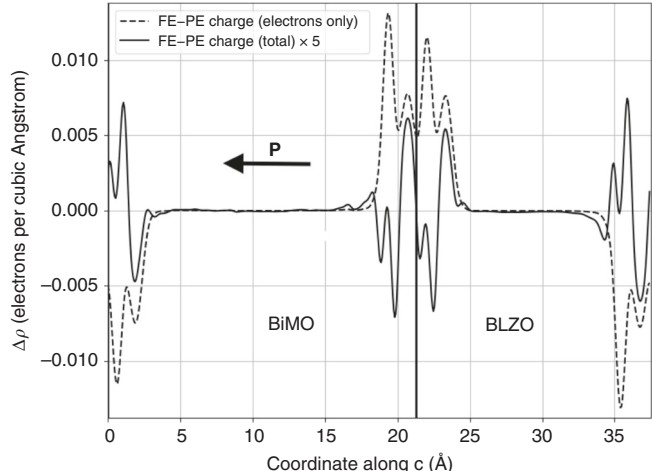

**Fig. 5 Interface charges in BiMO/insulator superlattice.** Filtered averages of electronic charge and of total charge in the BiMO–BLZO SL, showing polarization-originating charge accumulation at the interfaces. Solid line and $y$ axis mark the geometric interfaces. Negative charge drawn as positive.

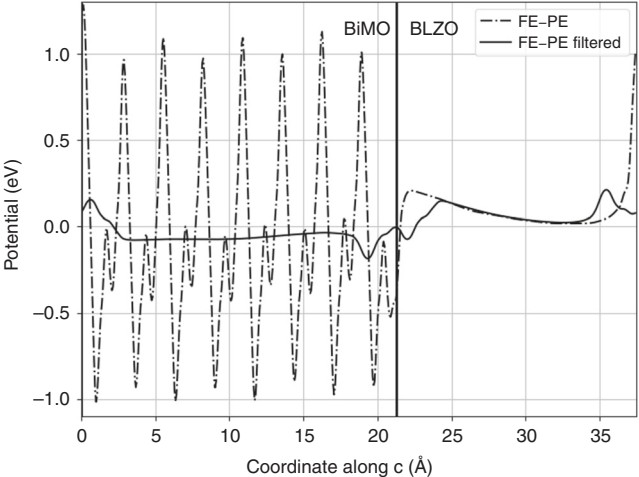

**Fig. 6 Electric fields in BiMO/insulator superlattice.** Filtered average of potential difference of polar-distorted and non-polar SLs, highlighting the sawtooth shape of the potential and the existence of a depolarizing field.

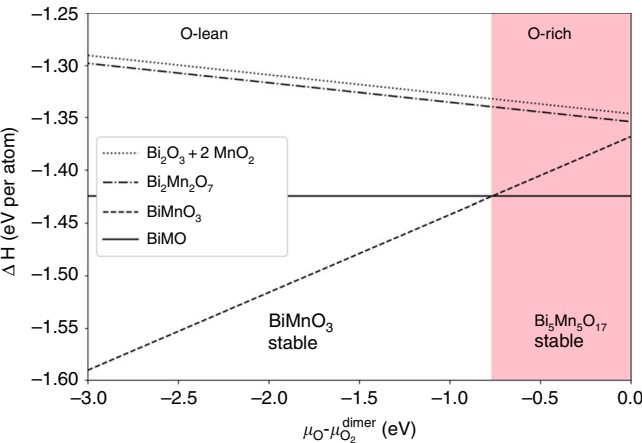

**Fig. 7 Thermodynamic stability of BiMO.** Formation enthalpies (from $O_2$, Bi and Mn metals) of BiMO, $BiMnO_3$, $Bi_2Mn_2O_7$, and $Bi_2O_3+2\,MnO_2$ vs O chemical potential. BiMO stability region: pink area on the right.

and $2 \times 10^{-8}$ in SI units, assuming the dielectric susceptibility $\chi_d \simeq 40\epsilon_0$ (see below) and a characteristically "large" magnetic susceptibility $\chi_m = 1000\,\mu_0$.

**Interface monopoles and depolarizing field.** The discontinuity of BiMO's zero-field **P** at the interface with an unpolarized insulating medium should produce[34] a sheet-like charge and hence a depolarizing field. If this were the case, BiMO's **P** would be switchable by an external field (see ref. [3]). This may be pre-empted, however, by conduction charge compensation, or by the disappearance of the polar distortion if the depolarizing field were too strong. Thus, to elucidate the possibility of switching BiMO's polarization, we study an insulator-cladded BiMO layer in a gated-device configuration.

Previously[3] we showed that $Bi_5Ti_5O_{17}$ supports a field when stacked with an insulator along its polar axis **b**; given the similarly flat Fermi surface of BiMO, we expect that state B will shadow that behavior closely, and we do not address it explicitly. We instead turn to state C, which is the harder case since, as **P**∥**c**, we need a metal/insulator superlattice (SL) along **c**, which requires a layered-perovskite $n = 5$ insulator (other claddings, including vacuum, would be highly prone to interface states). For our proof-of-concept simulations, we adopt the fictitious *Pmnn* compound $BaLa_4Zr_5O_{17}$ (BLZO), whose gap is 2 eV in GGA.

In Figs. 5 and 6 we report charge and potential differences between the non-polar *Pnnm* BiMO/*Pnnm* BLZO and polar *Pmn2₁* BiMO/*Pnnm* BLZO SLs (3/4 BiMO/BLZO cells, symmetric interfaces, 378 atoms). The macroscopically-averaged[3,35] charge difference shows a monopole, matching the polarization direction, that integrates to 0.19 $\mu$C cm$^{-2}$ (about 1/25 the bare **P**). This confirms that BiMO carries a non-zero **P** which the conduction charge is unable to screen out[3] (the conduction density of $3 \times 10^{21}$ cm$^{-3}$ is low, but still 20 times the needed screening density). The BiMO layer has a finite effective dielectric constant of $\varepsilon_{BiMO} \simeq 40$ (Eq. 4, ref. [34] with $\varepsilon = \varepsilon_\infty = 5$ for lattice-frozen BLZO), a suppression of the static Drude divergence being admissible from general features of metal–insulator SLs[36]. BLZO is non-polar and interfaces are symmetric, so the interface charge must stem from BiMO's polarization.

Accordingly, the polar–non-polar SL potential difference (Fig. 6) has a slope, i.e. an electric field inside both BiMO and the cladding, whose only possible source is the polarization within

the BiMO layer. The field is 190 MV m$^{-1}$ in the insulator; the field of 200 MV m$^{-1}$ from the interface monopole is in the same ballpark. Aside from its precise value, the depolarizing field confirms that the non-zero **P** of BiMO in state C survives as in the BiTO B-like state (which, we recall, BiMO also possesses in addition to the C-like state being discussed). Hence, BiMO qualifies as a multiferroic metal.

The screened field (energy density 1 $\mu$eV Å$^{-3}$) cannot remove the polar distortion (100 $\mu$eV Å$^{-3}$ energy density gain), whereas the unscreened field (energy density 700 $\mu$eV Å$^{-3}$) would. Thus, similarly to BiTO[3], we may label BiMO a self-screening hyperferroelectric magnetic metal, since polarization survives in the thin film thanks to self-screening (although of course the underlying mechanism in BiMO is quite unlike that in hyperferroelectrics proper[37]).

**Thermodynamic stability.** To assess BiMO's stability within equilibrium thermodynamics, in Fig. 7 we compare its enthalpy of formation (see Methods) with that of a few possible alternative Bi–Mn–O systems, specifically $BiMnO_3$ (a rare insulating ferromagnet, paraelectric in equilibrium, ferroelectric under strain[38]), $Bi_2Mn_2O_7$ (a layered-perovskite ferroelectric and antiferromagnetic insulator, not synthesized so far), and a combination of the two

binaries $Bi_2O_3$ and $MnO_2$ in their most stable versions vs the chemical potential of oxygen. $Mn_2BiO_5$, also considered, is not competitive. These Bi-Mn-O combinations are both oxygen-rich and oxygen deficient (with 3, 3.5, and 3.5 oxygen atoms per perovskite stoichiometric unit) compared BiMO (3.4 oxygens per unit). Clearly BiMO is the most stable of this group in an appreciable range of oxygen-rich conditions. While the stoichiometries considered here are not exhaustive, there is good circumstantial evidence for the possible stability of BiMO.

In summary, we have predicted that $Bi_5Mn_5O_{17}$ is a multiferroic metal featuring three space-orthogonal vector ferroic order parameters, and with two degenerate ground states where the order-parameter triad gets rotated in space. As such, the material is expected to exhibit multistate multiferroicity, nonreciprocity effects, and giant magnetoelectricity. Importantly, it has a thermodynamical stability window that should make it experimentally accessible. This material could be the realization of a long sought-after goal, a room-temperature single-phase multiferroic with large and strongly coupled polarization and magnetization.

## Methods

**Computational details**. First-principles density-functional calculations in the generalized gradient (GGA) and local density (LDA) approximations to density-functional theory are performed with VASP[39–42] and Quantum Espresso (QE)[43,44], and supplemented with variational pseudo-self-interaction-corrected (VPSIC) calculations[9,10]. Structural instabilities and magnetic properties are studied in GGA and LDA, and VPSIC is used for improved electronic structure and polarization properties. In the VPSIC code we use scalar-relativistic ultrasoft pseudopotentials[45] with plane-wave cutoff of 476 eV; in VASP we use scalar-relativistic projector augmented waves[46,47] (valence electrons: Bi $5d$, $6s$, $6p$; Mn $3d$, $4s$; O $2s$, $2p$; PAW data sets Bi, Mn, O_s) and a cutoff of 500 eV; in QE we use fully relativistic ultrasoft potentials with cutoff 90 Ry. Brillouin zone integration is done on a $6 \times 2 \times 4$ grid for self-consistency and optimization, and a $12 \times 4 \times 8$ grid to compute densities of states. The electronic polarization is computed with the Berry phase approach[48] as modified in ref. [3], on an $8 \times 4$ set of 11-point k-strings along the polarization axes (i.e. a reoriented $8 \times 4 \times 11$ grid). Non-collinear magnetism calculations, including spin–orbit effects, have been double checked with VASP and QE. Convergence tests have been performed for each code separately, and structural relaxations were redone independently with both. More information in the Supplementary Methods. Further details on the potential-filtering and polarization-calculation procedures are in ref. [3]. Formation enthalpy data (except the layered perovskites, which we computed directly) are from the Materials Project, https://materialsproject.org, with material IDs mp-23262 ($Bi_2O_3$), mp-19395 ($MnO_2$), mp-35 (Mn), mp-23152 (Bi); mp-504697 ($Mn_2BiO_5$), mp-23477 ($BiMnO_3$).

## Data availability

Input files and details of procedures can be provided upon request.

## Code availability

The main codes used are public domain (QE), licensed (VASP), or custom in-house (VPSIC). The latter can be provided in the frame of a scientific collaboration. Further minor postprocessing codes can be provided upon request.

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

## Acknowledgements

Work supported in part by UniCA, FdS, RAS via Progetti di ateneo 2016 *Multiphysics approach to thermoelectricity* (VF) and 2020 *Stability and defects of hybrid perovskites* (AF, VF); CINECA, Italy, via ISCRA grants (VF); Luxembourg National Research Fund, Grant No. INTER/ANR/16/11562984 EXPAND (JÍ); Project MIUR-PRIN 2017 TOP-SPIN (AF). VF is currently on secondment leave at the Embassy of Italy, Berlin, Germany; views expressed herein are his own and not necessarily shared by the Italian Ministry of Foreign Affairs.

## Author contributions

A.U., V.F., and J.I.: structure and its instabilies. A.F., A.U., F.R., and V.F.: electronic, magnetic, and transport properties. A.F.: polarization properties. A.U. and V.F.: magnetic non-collinear properties. F.R. and V.F.: behavior of finite layers and slabs. J.I. and V.F.: thermodynamic stability. A.U., J.I., and V.F.: Landau free energy and toroidicity. V.F.: main draft of the paper. A.U., A.F., F.R., J.I., V.F.: refinement and finalization of the paper.

## Competing interests

The authors declare no competing interests.
