## [Peer Review File · Nature Communications]

REVIEWER COMMENTS

Reviewer #1 (Remarks to the Author):

The mutual couplings of the various orders is very useful, unfortunately, such kind of materials are very rare. Thus to explore possible new candidate of multiferroic compounds is very important.

In this work, based on the density-function calculation, the authors predict that Bi₅Mn₅O₁₇ is a layered-perovskite metal and shows a ferromagnet, ferroelectric, and ferrotoroid. Thus they claim that this material is a single-phase multiferroic with large, strongly coupled, primary-order polarization and magnetization.

It is well known that the electronic correlation in 3d orbits is very important, and LDA+U method usually is more suitable for 3d system. Thus the authors should provide the LDA+U results and confirm their main results is not sensitive to that.

Reviewer #2 (Remarks to the Author):

Multiferroic materials, are interesting for technological application because they can combine in a single phase different orderings (e.g. ferroelectric, ferromagnetic). Most of known multiferroics crystallize in perovskite structures, and are usually ferroelectric (allowing the switching using the conjugate electric field) but antiferromagnetic. An interesting class of multiferroic materials are the magnetoelectric multiferroic materials, that allow a strong coupling between the different orders (magnetisation and polarization). In that context, the manuscript "A three-order-parameter, bistable, magnetoelectric multiferroic metal", presents a theoretical investigation of multiferroic BiMnO₃ in a different stoichiometry: Bi₅Mn₅O₁₇. They predict using DFT calculations, that this material should form in two close in energy ground states, where in both states the material would be Ferroelectric (although metallic), Ferromagnetic and conclude that by symmetry it should also be Ferrotoroid, making it a metallic multiferroic. In both ground states, the three order parameters are orthogonal along different directions, depending on the ground state. This implies a strong magnetoelectric effect by going from one ground state to the other one. Finally, the authors give some thermodynamical consideration to show the possibility for the (5,5,17) stoichiometry to be more stable than the (1,1,3) stoichiometry under O-rich conditions.

The manuscript is well written and is certainly of interest for the multiferroic community (at least). I read the previous work of the authors on the ferroelectric metal (Nature Commun. 7, 11211 (2016)), but would still be somehow more careful on the strong affirmation of the discovery of a new Ferroelectric metal, maybe a "polar metal possibly ferroelectric" would be safer until it has experimentally been shown that one can indeed switch the polarization in such systems.

On a more technical level I would like the authors to clarify the following points:

The authors write that they checked their non-collinear magnetism calculations in VASP and Quantum Espresso; they don't specify if the structure was previously optimized in each code or only relaxed in one code and transferred to the other code. For the sake of clarity, it should be mentioned.

When presenting the two different ground states, the authors explain that these structures were found by looking at instabilities in the phonon spectrum. How were the phonons computed? did the authors use a frozen phonon technique or DFPT?

Still concerning the structures, could the authors give more precision on the different distortions present in each structure with respect to the higher symmetry structure? I understand that both structure share the same space group but with different orientations, although I struggle understanding the different distortions and the arrows on figure1 don't really help. What are the lattice

vectors in State C and State B? are they identical?

It is claimed that it is possible to go from one ground state to the other one by changing the temperature to over go the energy barrier between both structures. The energy landscape is an upper bound estimation as it was (probably?) computed by only freezing the distortions. This is an open question: would a NEB calculation between both structures have a chance to reveal another hidden ground state?

Concerning the ferroelectric well depth, it is claimed that the depth is similar to that of BaTiO₃. How was the double well calculated? Could the author present the double well calculation going from the polar structure to its inverse?

The system was found to be Ferromagnetic and the energy difference between FM and AFM G-type was used to compute the exchange energy, but were other magnetic ordering tested to verify that the ground states are FM ?

The symmetry analysis gets to the conclusion that there should exist a toroidal moment perpendicular to both other order parameters. In general, the toroidal moment is only defined for non-magnetic system (no net magnetisation). I am not expert in this field but believe that the authors are carrying a calculation that is correct. Although, I suggest that they could clarify the reason why their calculation is justified in a maybe more explicit way for non-experts.

Finally, a more open question: do the authors have suggestions on how one could target one or the other ground state? Could there be some clever choice of substrate to force the orientation?

Reviewer #3 (Remarks to the Author):

The manuscript from Urru and co-workers entitled " A three-order-parameter, bistable, magnetoelectric multiferroic metal" presents a rather meticulous study of the electronic properties of the orthorhombic layered perovskite Bi₅Mn₅O₁₇. This is found to be a metallic magnetic ferroelectric, displaying also a ferrotoroidal moment. The three order parameters are coupled to each other so that this has to be considered as a true multiferroic material. Furthermore, estimates of the critical temperatures for both magnetism and ferroelectricity suggest that such multiferroic state can be observable at room temperature. The paper is entirely theoretical and such materials is here proposed, but has never been synthesised in the lab. I am not an expert in this field and for me it is difficult to evaluate both the novelty and the potential appeal of the claims made here. As such I will limit my report to comments only on technical aspects of the manuscript. From the technical point of view, I believe that this is a thorough piece of work, which does not seem to present any particular issues. I believe that there are several points that the authors have to clarify, but in general this is a solid work.

The authors have to reflect on the following points:

- 1) At page two it is said that there is an energy barrier of approximately 50 $\mu\text{eV}/\text{\AA}^3$ between the two quasi-degenerate ferroelectric ground states. From this value the authors extract a critical temperature of about 400K. I am not sure I follow this argument. The energy barrier between the two ground states appear to me more as the energy associated to a coherent transition between such two states. This would determine the equivalent of a "blocking temperature", but I don't understand how this single number can be related to the critical temperature. It appears to me that it is the phonon spectrum of the material that eventually dictates the transition temperature, not just the energy barrier between degenerate ground states.

2) A somewhat similar analysis is done for the magnetic critical temperature. This is estimated by taking the energy difference between the ferromagnetic and an antiferromagnetic phase and by fitting such difference to a nearest-neighbour Heisenberg model. Such procedure gives an exchange parameter, which is then used in the expression for the mean field critical temperature (I guess this is what is done, since the authors don't really explain it). Such procedure has a number of issues: 1) giving the fact that the material is a metal, it is likely the the range of the interaction extends well beyond nearest neighbours; 2) some of the exchange parameters beyond nearest neighbours can be negative and the overall critical temperature may be sensibly different from what is calculated here; 3) the mean field expression almost always overestimate the critical temperature. Demonstrating that Bi₅Mn₅O₁₇ is a multiferroic above room temperature is a key message of the manuscript, so the estimates of the critical temperatures need to be done more carefully.

3) I do not think that the authors have really demonstrated that the material is thermodynamically stable. A careful analysis should explore whether the compound is robust against decomposition along competing binary and ternary compounds. I do not know how many ternaries exist for this chemical composition, in addition to BiMnO₃, but certainly there are several composition pathways that need to be checked (involving BiMnO₃ and some binaries). A through analysis would consider decomposition along all possible binaries (including hypothetical). This is not what I am suggesting here, but some likely decomposition paths need to be checked. At the end of the day it does not matter what are the properties of a compound if this does not exist.

In conclusion this is an interesting piece of work. I am not in the position to comment on the novelty and appeal, although to my non-expert eyes it looks interesting. From the technical side there are three claims, pretty fundamental to make a case, which are not fully supported by the theoretical evidence. These are related to the two critical temperature and to whether the material can be made. Such issues need to be addressed.

REVIEWER COMMENTS

Reviewer #1 (Remarks to the Author):

The mutual couplings of the various orders is very useful, unfortunately, such kind of materials are very rare. Thus to explore possible new candidate of multiferroic compounds is very important.

In this work, based on the density-function calculation, the authors predict that Bi₅Mn₅O₁₇ is a layered-perovskite metal and shows a ferromagnet, ferroelectric, and ferrotoroid. Thus they claim that this material is a single-phase multiferroic with large, strongly coupled, primary-order polarization and magnetization.

It is well known that the electronic correlation in 3d orbits is very important, and LDA+U method usually is more suitable for 3d system. Thus the authors should provide the LDA+U results and confirm their main results is not sensitive to that.

#####

Reply to Reviewer 1

We thank Reviewer #1 for sharing the concern about possible correlation effects in the 3d shell of Mn in our material, and suggesting LDA+U (or similar) to countercheck our results for magnetism.

We performed calculations of the energy difference of FM and AF-G in LDA+U as function of U, and report them in the picture attached. (We also tried to calculate AF-C and AF-A but ended up with non-magnetic states.) The system remains robustly FM, with no major changes.

Interestingly, the calculations show (figure below) that the J coupling parameter increases with U (i.e. FM gets stronger), and at the typical U = 3 eV for Mn it is about 50% larger than at U=0. This reinforces our prediction of above room-temperature magnetism. This is expected in retrospect, as U should enlarge the eg-t_{2g} energy distance, reducing the AFM contribution to the exchange from filled t_{2g} (a typical LDA or GGA problem is indeed to overestimate AF magnetism as a consequence of having t_{2g} too high in energy).

As a general comment, beyond-LDA effects (significant in insulating manganese oxides such as LaMnO₃ where magnetism is dominated by superexchange) are only moderately important in materials such as BiMO which are effectively n-doped and therefore metallic manganites, so that FM is expected from the dominance of magnetic double exchange in metals, similarly to La(x)Sr(1-x)MnO₃ (see our previous work in Phys.Rev.B 76, 064428 (2007); 78, 235122 (2008); Eur.Phys.J.B 70, 343 (2009); PRB 82, 140101(R) (2010)). Further, the bands in Fig. 3 were already calculated with a beyond-LDA method, known to perform well for correlated materials (a review is in Eur.Phys.J.B 71, 139 (2009)).

Reviewer #2 (Remarks to the Author):

Multiferroic materials, are interesting for technological application because they can combine in a single phase different orderings (e.g. ferroelectric, ferromagnetic). Most of know multiferroics crystallize in perovskite structures, and are usually ferroelectric (allowing the switching using the conjugate electric field) but antiferromagnetic. An interesting class of multiferroic materials are the magnetoelectric multiferroic materials, that allow a strong coupling between the different orders (magnetisation and polarization). In that context, the manuscript “A three-order-parameter, bistable, magnetoelectric multiferroic metal”, presents a theoretical investigation of multiferroic BiMnO₃ in a different stoichiometry: Bi₅Mn₅O₁₇. They predict using DFT calculations, that this material should form in two close in energy ground states, where in both states the material would be Ferroelectric (although metallic), Ferromagnetic and conclude that by symmetry it should also be Ferrotoroid, making it a metallic multiferroic. In both ground states, the three order parameters are orthogonal along different directions, depending on the ground state. This implies a strong magnetoelectric effect by going from one ground state to the other one. Finally, the authors give some thermodynamical consideration to show the possibility for the (5,5,17) stoichiometry to be more stable than the (1,1,3) stoichiometry under O-rich conditions.

The manuscript is well written and is certainly of interest for the multiferroic community (at least). I read the previous work of the authors on the ferroelectric metal (Nature Commun. 7, 11211 (2016)), but would still be somehow more careful on the strong affirmation of the discovery of a new Ferroelectric metal, maybe a “polar metal possibly ferroelectric” would be safer until it has experimentally been shown that one can indeed switch the polarization in such systems.

On a more technical level I would like the authors to clarify the following points:

The authors write that they checked their non-collinear magnetism calculations in VASP and Quantum Espresso; they don't specify if the structure was previously optimized in each code or only relaxed in one code and transferred to the other code. For the sake of clarity, it should be mentioned.

When presenting the two different ground states, the authors explain that these structures were found by looking at instabilities in the phonon spectrum. How were the phonons computed? did the authors use a frozen phonon technique or DFPT?

Still concerning the structures, could the authors give more precision on the different distortions present in each structure with respect to the higher symmetry structure? I understand that both structure share the same space group but with different orientations, although I struggle understanding the different

distortions and the arrows on figure1 don't really help. What are the lattice vectors in State C and State B? are they identical?

It is claimed that it is possible to go from one ground state to the other one by changing the temperature to over go the energy barrier between both structures. The energy landscape is an upper bound estimation as it was (probably?) computed by only freezing the distortions. This is an open question: would a NEB calculation between both structures have a chance to reveal another hidden ground state?

Concerning the ferroelectric well depth, it is claimed that the depth is similar to that of BaTiO₃. How was the double well calculated? Could the author present the double well calculation going from the polar structure to its inverse?

The system was found to be Ferromagnetic and the energy difference between FM and AFM G-type was used to compute the exchange energy, but were other magnetic ordering tested to verify that the ground states are FM ?

The symmetry analysis gets to the conclusion that there should exist a toroidal moment perpendicular to both other order parameters. In general, the toroidal moment is only defined for non-magnetic system (no net magnetisation). I am not expert in this field but believe that the authors are carrying a calculation that is correct. Although, I suggest that they could clarify the reason why their calculation is justified in a maybe more explicit way for non-experts.

Finally, a more open question: do the authors have suggestions on how one could target one or the other ground state? Could there be some clever choice of substrate to force the orientation?

==== Reply to Reviewer #2

We thank the referee for the assessment. We copy and paste the main points to reply to them point-by-point below.

Rev 2 #1 =====

On a more technical level I would like the authors to clarify the following points:

The authors write that they checked their non-collinear magnetism calculations in VASP and Quantum Espresso; they don't specify if the structure was previously optimized in each code or only relaxed in one code and transferred to the other code. For the sake of clarity, it should be mentioned.

Reply =====

The original analysis based on VASP calculations has been redone entirely with QE reoptimizing everything, with completely consistent results. We mention this in the revised version of the manuscript.

Rev 2 #2 =====

When presenting the two different ground states, the authors explain that these structures were found by looking at instabilities in the phonon spectrum. How were the phonons computed? did the authors use a frozen phonon technique or DFPT?

Reply =====

The phonon modes at $q=0$ of parent phase Pmnn were obtained with the DFPT approach using Quantum Espresso. Earlier on, we had computed the modes by finite differences with VASP, with completely consistent results; QE offers in addition a detailed symmetry analysis.

The two "most unstable" modes had $f/i \sim 100 \text{ cm}^{-1}$, and two more were less unstable at $f/i \sim 40 \text{ cm}^{-1}$. A symmetry analysis of the modes shows them to be polar distortions with irreps B1u, B2u, B3u, Au. The "most unstable" modes (B1u, B3u) lead to distorted structures with space groups Pmn21 and Pm21n, respectively, and end up being almost degenerate (these are the C and B states discussed in the ms). The B2u and Au lead to distorted structures with groups P21mn and P21212, respectively, which are higher in energy. We added a sentence in the revised text to clarify how phonons were computed.

Rev 2 #3 =====

Still concerning the structures, could the authors give more precision on the different distortions present in each structure with respect to the higher symmetry structure? I understand that both structures share the same space group but with different orientations, although I struggle understanding the different distortions and the arrows on figure 1 don't really help. What are the lattice vectors in State C and State B? are they identical?

Reply =====

The lattice vectors (Fig. 1, bottom left) are the same for all three structures; same lattice parameters for the two distorted states B and C, different lattice parameters in the parent structure (top left, Fig. 1). As mentioned in the previous point, states C and B are generated by distortions associated with irreducible representations B1u and B3u, respectively. Hence, it is important to emphasize that these two structures are not related by symmetry; the fact that they are nearly degenerate in energy is a by-product of the similarity of the corresponding distortions, but it is not forced by symmetry. (Interestingly, the distortions would be related by symmetry if we had a perfect three-dimensional perovskite lattice.) 4 operations are broken in each of the structures, but not the same 4. In particular, in C the mirror normal to c is broken, in B the mirror normal to b. The arrows in Fig. 1 center and right are schematic and not to scale, and positioned in the approximate location of the largest dipoles (there are many smaller and hardly readable one). The latter dipoles correspond to changes in Bi-O bond length.

Rev 2 #4 =====

It is claimed that it is possible to go from one ground state to the other one by changing the temperature to overcome the energy barrier between both structures. The energy landscape is an upper bound estimation as it was (probably?) computed by only freezing the distortions. This is an open question: would a NEB calculation between both structures have a chance to reveal another hidden ground state?

Reply =====

The landscape is indeed mapped out by frozen distortions. Given the symmetry analysis, any accessible distortion should be a combination of the two B3u and B1u modes, which rules out an alternate ground state. Put differently, it is exceedingly unlikely that a NEB calculation could result in an alternative ground state combining B1u and B3u features; if such a state existed, we would expect to see indications of it (such as a region with relatively low energy away from the B3u and B1u minima) in our frozen-distortion simulations, but this is not the case. This is confirmed by a further calculation, namely the relaxation under the lowest possible symmetry: the resulting structure, which has group P21 and is a combination of all unstable modes, also ends up being higher in energy than our B and C states.

Rev 2 #5 =====

Concerning the ferroelectric well depth, it is claimed that the depth is similar to that of BaTiO₃. How was the double well calculated? Could the author present the double well calculation going from the polar structure to its inverse?

Reply =====

We are not sure we understand which double wells the referee is talking about, the ones for Bi₅Mn₅O₁₇ or those of BaTiO₃. Nevertheless, let us explain our argument, hoping that will clarify the situation.

For Bi₅Mn₅O₁₇, the results are those of Fig. 2 in our manuscript, and the calculations are described in our text. The depth of the ferroelectric wells, as measured from the para electric phase, is about 100 microeV/A³ for the two polar states discussed in our paper. As for the barrier between these two states, we estimate it to be of about 50 microeV/A³.

As regards BaTiO₃, this is a very well-known compound and a lot of information is available in the literature. For example, from the work of King-Smith and Vanderbilt [Phys. Rev. B 49, 5828 (1994)], we can deduce that the energy difference between the paraelectric cubic phase and the polar rhombohedral ground state is about 240 microeV/A³. We can also estimate the energy barrier between neighbouring rhombohedral phases with polarisations along (1,1,1) and (1,1,-1) directions, respectively, by inspecting the energy of the orthorhombic saddle point, with polarisation along (1,1,0), that serves as a bridge for the transition from one to the other. According to the data provided by King-Smith and Vanderbilt, this energy barrier is about 15 microeV/A³ in BaTiO₃. These energy differences are quite comparable to those obtained for Bi₅Mn₅O₁₇, the barriers between competing polar orders being actually higher for the compound of interest here.

Now, there is theoretical evidence that structural transition temperatures in perovskite oxides and related materials are roughly proportional to the barriers that need to be overcome to escape local minima in the energy landscape [see e.g. Wojdel and Íñiguez, Phys. Rev. B 90, 014105 (2014)]. This relationship has also been established empirically, although in the experiments attention is paid to the (square of) the distortion associated to the symmetry breaking transition, which, in turn, can be seen to be proportional to the relevant energy barrier in most typical cases [Abrahams et al., Phys. Rev. 172, 551 (1968)]. Having this in mind, we note that transition temperature for the orthorhombic-to-rhombohedral phase transition in BaTiO₃ is 183 K. This temperature corresponds to a computed energy barrier of 15 microeV/A³ to escape from the rhombohedral well. Hence, we can roughly estimate that in Bi₅Mn₅O₁₇, where the energy needed to escape is about 3 times higher, the ordering transition can be expected to be well above room temperature.

Admittedly, this is not a definitive argument, as a rigorous determination of the transition temperature would require a statistical simulation of Bi₅Mn₅O₁₇, which is not feasible computationally. Nevertheless, we think our computed energy differences support the notion that, indeed, this compound will be ferroelectric at temperatures well above room.

We have extended this discussion in the manuscript, to better explain our claim of room-temperature ferroelectricity.

Rev 2 #6 =====

The system was found to be Ferromagnetic and the energy difference between FM and AFM G-type was used to compute the exchange energy, but were other magnetic ordering tested to verify that the ground states are FM ?

Reply =====

Yes, we looked at a number of AF versions in the insulating relative Bi₂Mn₂O₇, and brought those over to BiMO. AF-G is lowest, as in fact other AF orders (A, C) are not stable solutions. FM is favored overall, as expected from Goodenough-Kanamori-Anderson double exchange.

Rev 2 #7 =====

The symmetry analysis gets to the conclusion that there should exist a toroidal moment perpendicular to both other order parameters. In general, the toroidal moment is only defined for non-magnetic system (no net magnetisation). I am not expert in this field but believe that the authors are carrying a calculation that is correct. Although, I suggest that they could clarify the reason why their calculation is justified in a maybe more explicit way for non-experts.

Reply =====

We may be missing the implications of the question, but the natural definition of toroidicity we refer to is that of magnetotoroidicity, and is valid for magnetic systems. There exists an electrotoroidicity (toroidicity for dipolar systems), which does not exist in our system by symmetry. Further, in principle, a magnetotoroidal moment may or may not coexist with a net magnetization (i.e. the system could be ferro or ferri or antiferromagnetic. A magnetic toroidicity can even be generated by an appropriate current density (but this is outside of our present scope).

Rev 2 #8 =====

Finally, a more open question: do the authors have suggestions on how one could target one or the other ground state? Could there be some clever choice of substrate to force the orientation?

Reply =====

We suggest in the text that an interesting route would be to convert B into C and viceversa by concurrent thermal and field driving (because e.g. $M \parallel a$ in B, and $M \parallel b$ in C). We have now also added a discussion of the internal degeneracy of both C and B. Aside from that, of course one could explore a whole world of further possibilities applying a strain (epitaxial or otherwise), for example.

=====

Reviewer #3 (Remarks to the Author):

The manuscript from Urru and co-workers entitled " A three-order-parameter, bistable, magnetoelectric multiferroic metal" presents a rather meticulous study of the electronic properties of the orthorhombic layered perovskite Bi₅Mn₅O₁₇. This is found to be a metallic magnetic ferroelectric, displaying also a ferrotoroidal moment. The three order parameters are coupled to each other so that this has to be considered as a true multiferroic material. Furthermore, estimates of the critical temperatures for both magnetism and ferroelectricity suggest that such multiferroic state can be observable at room temperature. The paper is entirely theoretical and such materials is here proposed, but has never been synthesised in the lab. I am not an expert in this field and for me it is difficult to evaluate both the novelty and the potential appeal of the claims made here. As such I will limit my report to comments only on technical aspects of the manuscript. From the technical point of view, I believe that this is a thorough piece of work, which does not seem to present any particular issues. I believe that there are several points that the authors have to clarify, but in general this is a solid work.

The authors have to reflect on the following points:

==== Reply to Reviewer 3

We thank the referee for the assessment. We copy and paste the main points to reply to them point-by-point below.

Rev 3 #1 =====

1) At page two it is said that there is an energy barrier of approximately $50 \mu\text{eV}/\text{\AA}^3$ between the two quasi-degenerate ferroelectric ground states. From this value the authors extract a critical temperature of about 400K. I am not sure I follow this argument. The energy barrier between the two ground states appear to me more as the energy associated to a coherent transition between such two states. This would determine the equivalent of a "blocking temperature", but I don't understand how this single number can be related to the critical temperature. It appears to me that it is the phonon spectrum of the material that eventually dictates the transition temperature, not just the energy barrier between degenerate ground states.

==== Reply

This is a fair comment, essentially identical to comment #5 made by Reviewer #2. We refer the reviewer to our detailed response above.

The reviewer's reference to phonons, if we interpret it correctly, is indeed correct: transition probabilities (as e.g. in the Vineyard-Arrhenius formula $P \sim f \exp(-dE/kT)$) do depend on the attempt (i.e. average phonon) frequency f . However the dependence is only linear, whereas that on the energy barrier dE is exponential.

Rev 3 #2 =====

2) A somewhat similar analysis is done for the magnetic critical temperature. This is estimated by taking the energy difference between the ferromagnetic and an antiferromagnetic phase and by fitting such difference to a nearest-neighbour Heisenberg model. Such procedure gives an exchange parameter, which is then used in the expression for the mean field critical temperature (I guess this is what is done, since the authors don't really explain it). Such procedure has a number of issues: 1) giving the fact that the material is a metal, it is likely the the range of the interaction extends well beyond nearest neighbours; 2) some of the exchange parameters beyond nearest neighbours can be negative and the overall critical temperature may be sensibly different from what is calculated here; 3) the mean field expression almost always overestimate the critical temperature. Demonstrating that Bi₅Mn₅O₁₇ is a multiferroic above room temperature is a key message of the manuscript, so the estimates of the critical temperatures need to be done more carefully.

==== Reply

Estimating T_c is of course daunting in general. We extracted J from the FM-AFG difference and plugged it into expressions for T_c obtained in the literature from simulations of the Ising and Heisenberg models (Ising formula, 3D cubic lattice; quantum-Heisenberg-model high-temperature-expansion numerical result for the spin 3/2 FM cubic lattice). Hence, we did not resort to any mean-field approximation, but to numerical solutions of model systems existing in the literature. In the new version, we revised all the data and formulas, and added the classical Heisenberg model result. The table below summarizes T_c in K:

	H-c	H-q	Ising
U=0:	361	263	378
U=3:	531	387	555

We include two sets of values obtained from +U calculations performed in response to reviewer #1, namely with U=0 and the typical U=3 of manganites. With large uncertainties, it looks like a fair guess is that T_c is above room temperature.

As regards the other concerns of the Reviewer, it is certainly true that multiple J, both FM and AF, are possible in general. When applicable they can be extracted (e.g. our PRL 95, 086405 (2005) and PRB 86, 195132 (2012)) combining energies of “magnetic excitations” such as FM, AF-G, -A, -C, etc. In our case, however, all AF states except the G type converge to non-magnetic solutions, suggesting that longer-range J couplings are not relevant. More generally, this is likely due to BiMO being effectively an n-doped (and therefore metallic) manganite at doping 0.2 from the nearest insulating phase, and therefore robustly FM as expected from double exchange, similarly to La(x)Sr(1-x)MnO₃ (see our previous work in Phys.Rev.B 76, 064428 (2007); 78, 235122 (2008); Eur.Phys.J.B 70, 343 (2009); PRB 82, 140101(R) (2010)).

Concerning long-range free-electron-gas-like interactions, we suspect from the Fermi surface that they will be unimportant along the b* and c* reciprocal-space axes, due to the very flat bands; the Fermi surface sheet-like shape across the a* direction, on the other hand, suggest a potential for developing a long-wavelength (~50 Å or so) spin-wave order, which would probably be incommensurate, and hence unassailable computationally.

We have revised the manuscript to mention the essence of the discussions above.

Rev 3 #3 =====

3) I do not think that the authors have really demonstrated that the material is thermodynamically stable. A careful analysis should explore whether the compound is robust against decomposition along competing binary and ternary compounds. I do not know how many ternaries exist for this chemical composition, in addition to BiMnO₃, but certainly there are several composition pathways that need to be checked (involving BiMnO₃ and some binaries). A through analysis would consider decomposition along all possible binaries (including hypothetical). This is not what I am suggesting here, but some likely decomposition paths need to be checked. At the end of the day it does not matter what are the properties of a compound if this does not exist.

===== Reply

As suggested, we considered a few additional decomposition routes into Bi and/or Mn oxides that appeared relevant from their formation energies, comparing BiMO with Bi₂Mn₂O₇ (a close relative layered perovskite), Bi₂O₃ plus MnO₂ (assuming the most stable of their many variants), and Mn₂BiO₅, in addition to BiMnO₃. Our previous conclusions are confirmed as shown in the new version of the figure, below (Mn₂BiO₅ is not reported as it is high outside the figure).

REVIEWER COMMENTS

Reviewer #1 (Remarks to the Author):

Carefully reading the response, I think the authors answer my comment and make modification at the SM. I now have no further questions.

Reviewer #2 (Remarks to the Author):

I thank the authors for the clear point by point answer and feel that they seriously answered my questions. I got in particular a better understanding on the magnetotoroidicity.

They also added more details and discussion in the manuscript accordingly.

I don't have any other questions/remarks and would recommend the manuscript "A three-order-parameter, bistable, magnetoelectric multiferroic metal" for publication.